# Characteristics and limitations of national antimicrobial surveillance according to sales and claims data

Yoshiki Kusama[1,2]*, Yuichi Muraki[3], Chika Tanaka[1], Ryuji Koizumi[1], Masahiro Ishikane[1,4], Daisuke Yamasaki[5], Masaki Tanabe[5], Norio Ohmagari[1,2,4]

**1** AMR Clinical Reference Center, Disease Control and Prevention Center, National Center for Global Health and Medicine, Tokyo, Japan, **2** Collaborative Chairs Emerging and Reemerging Infectious Diseases, National Center for Global Health and Medicine, Graduate School of Medicine, Tohoku University, Sendai, Miyagi, Japan, **3** Department of Clinical Pharmacoepidemiology, Kyoto Pharmaceutical University, Kyoto, Japan, **4** Disease Control and Prevention Center, National Center for Global Health and Medicine, Tokyo, Japan, **5** Department of Infection Control and Prevention, Mie University Hospital, Tsu, Mie, Japan

* stone.bagle@gmail.com

## Abstract

### Purpose

Antimicrobial use (AMU) is estimated at the national level by using sales data (S-AMU) or insurance claims data (C-AMU). However, these data might be biased by generic drugs that are not sold through wholesalers (direct sales) and therefore not recorded in sales databases, or by claims that are not submitted electronically and therefore not stored in claims databases. We evaluated these effects by comparing S-AMU and C-AMU to ascertain the characteristics and limitations of each kind of data. We also evaluated the interchangeability of these data by assessing their relationship.

### Methods

We calculated monthly defined daily doses per 1,000 inhabitants per day (DID) using sales and claims data from 2013 to 2017. To assess the effects of non-electronic claim submissions on C-AMU, we evaluated trends in the S-AMU/C-AMU ratio (SCR). To assess the effects of direct sales of S-AMU, we divided AMU into generic and branded drugs and evaluated each SCR in terms of oral versus parenteral drugs. To assess the relationship between S-AMU and C-AMU, we created a linear regression and evaluated its coefficient.

### Results

Median annual SCRs from 2013 to 2017 were 1.046, 0.993, 0.980, 0.987, and 0.967, respectively. SCRs dropped from 2013 to 2015, and then stabilized. Differences in SCRs between branded and generic drugs were significant for oral drugs (0.820 vs 1.079) but not parenteral drugs (1.200 vs 1.165), suggesting that direct sales of oral generic drugs were omitted in S-AMU. Coefficients of DID between S-AMU and C-AMU were high (generic, 0.90; branded, 0.84) in oral drugs but relatively low (generic, 0.32; branded, 0.52) in parenteral drugs.

**Data Availability Statement:** Researchers should apply directly to IQVIA Japan (https://www.iqvia.com/ja-jp/locations/japan) for access to the study's antimicrobial drug sales data. The use of the

criteria described in the Methods section will enable other researchers to identify the same data obtained for this study. The authors confirm that they did not have any special privileges to access the data that other researchers would not have. For the insurance claims data, the Ministry of Health, Labour and Welfare of Japan has placed strict legal restrictions on the release or sharing of these data. As a result, these data are not publicly available. But researchers can apply for such data as we used to the Ministry of Health, Labour and Welfare of Japan if passed qualification examination (phone number: +81-50-5546-9167).

**Funding:** This study was supported by a research grant from the Ministry of Health, Labour and Welfare of Japan (20HA2003).

**Competing interests:** The authors have declared that no competing interests exist.

## Conclusions

The omission of direct sales information and non-electronically submitted claims have influenced S-AMU and C-AMU information, respectively. However, these data were well-correlated, and it is considered that both kinds of data are useful depending on the situation.

## Introduction

Antimicrobial resistance has rapidly become a global public health and economic concern, and a substantial reduction in antimicrobial use (AMU) is needed to suppress the spread of resistant pathogens [1]. To monitor current usage patterns and support the implementation of effective antimicrobial stewardship programs, countries must first establish rigorous national AMU surveillance systems [2–4].

AMU can be estimated from antimicrobial drug sales data (S-AMU) or insurance claims data (C-AMU). S-AMU can be quickly and easily accessed and therefore is widely used in both developed and developing countries [5]. In contrast, C-AMU presupposes a sophisticated health insurance system and requires claims data to be collected electronically; therefore mainly developed countries use claims data to calculate national AMU [6]. A Europe-wide network of national AMU surveillance systems found that 15 countries used S-AMU, 4 used C-AMU, and 9 used a combination of the two in 2018 [7].

Understanding the limitations of each method is important in the monitoring of national AMU. Although data coverage is a key factor in the validity of S-AMU, data on antimicrobials that are not sold by wholesalers (i.e., those sold directly by pharmaceutical companies to medical facilities) are missed because the S-AMU dataset is compiled using data from wholesalers in Japan. Domestically, direct sales are more common for generic drugs than for branded drugs.

For C-AMU, the scope of medical care covered by health insurance is a key factor in the validity of the dataset. In Japan, nearly all medical care is covered by the national health insurance system, with a few exceptions such as cosmetic surgery and travel medicine (the number of items covered has not changed over time, according to our study conducted from 2011 to 2013 [8]). In addition, the prevalence of electronic claims submission systems also affects data coverage because paper claims are not recorded in the government-managed database. In Japan, such systems were not prevalent until 2015, and thus the coverage of claims recorded in the database before 2015 is incomplete [9].

AMU surveillance was initiated using S-AMU, but currently, the government-managed claims database can be accessed only by applicants who have pass the qualification exam [10]. We obtained the claims data, and evaluated the characteristics and limitations of national AMU according to sales and claims data, using S-AMU and C-AMU in Japan.

## Methods

### Study design

This retrospective database study was performed to examine monthly trends in S-AMU and C-AMU and to evaluate the characteristics and limitations of these data.

### Data sources

**S-AMU.** We estimated S-AMU by using commercial data purchased from IQVIA Japan, a private data firm that curates databases of medical drug sales and distribution throughout the

country. In Japan, medical drugs are generally sold by pharmaceutical companies to wholesalers, who then sell them to medical facilities [11–13]. IQVIA Japan collects sales data from these wholesalers in order to construct the databases. Although detailed information has not been published, the company states that it collects data from nearly all domestic wholesalers, and that the data encompass more than 99% (monetary value) of all wholesale drug sales in Japan (this coverage was confirmed through personal communication with IQVIA Japan). Our dataset contains wholesalers' sales data on all antimicrobials, as well as information on oral/parenteral classification and branded/generic classification. However, the S-AMU dataset does not include information on the purchasing parties (i.e., names and types of medical facilities) or patients, and therefore does not allow distinctions to be made between primary care and hospital care dispensing and reimbursement. In addition, although geographic information was available, we did not use it because of uncertainty about its validity.

**C-AMU.** We estimated C-AMU by using data obtained from the National Database of Health Insurance Claims and Specific Health Checkups of Japan (NDB), which is managed by the Ministry of Health, Labour and Welfare (MHLW). The NDB contains national-level health insurance claims data for healthcare encounters covered by insurance, and these data can be used for research purposes following the submission and approval of an application to the government [10]. Our dataset includes data on the dates and types of antimicrobials prescribed to patients for all insurance-covered healthcare encounters.

In Japan, all residents are required to enroll in health insurance, which entitles them to receive insurance-covered healthcare at any medical facility throughout the country. Enrollees pay monthly premiums to their insurers, and also pay a fixed proportion (10%–30% depending on age and income) of the medical charges at the point of care. Healthcare providers send claims to the applicable insurers to be reimbursed for the remaining charges. Because insurance-covered care accounts for the majority of medical treatments provided in Japan, the NDB represents a near-comprehensive database of all treatments performed throughout the country [8, 14].

However, the NDB does not include claims data from patients with fully publicly funded healthcare (e.g., patients with intractable diseases, atomic bomb survivors, patients on welfare, patients with tuberculosis, and patients with human immunodeficiency virus infections) and patients who personally pay for all of their medical expenses (e.g., international travelers and cosmetic surgery patients).

Furthermore, the NDB includes only those data from claims that have been submitted electronically, and the lack of paper claims reduces its coverage of healthcare encounters. Under Japan's healthcare system, claims for medical care, dental care, and drug dispensing (by pharmacies) are handled separately. Since 2011, electronically submitted claims for hospital-based medical care and drug dispensing claims have accounted for over 99.9% of all claims [9]. For clinic-based medical care, the proportion of electronically submitted claims increased from 91.0% in 2011 to 99.9% in 2015. For dental care, the proportion of electronically submitted claims have increased dramatically in recent years, from 31.5% in 2011 to 96.0% in 2015. Therefore, most claims have been submitted electronically since 2015, meaning the NDB has recorded nearly all medical, dispensing, and dental information since then.

## AMU measurement

We calculated the monthly S-AMU and C-AMU from January 2013 to December 2017. Antimicrobials are identified using the Anatomical Therapeutic Chemical (ATC) classification J01 (anti-infectives for systemic use), and the various drug categories were analyzed according to their respective ATC codes [15]. Route of administration (oral or parenteral) was also

categorized according to the ATC classification. AMU was calculated using defined daily dose (DDD) per 1,000 inhabitants per day (DID). This indicator, which shows a drug's assumed average maintenance dose (i.e., not loading dose) per day, was developed by the World Health Organization Collaborating Centre for Drug Statistics Methodology [15]. DID was calculated using the following formula.

$$\text{DID} = \frac{\text{Use in each month (g)}}{\text{DDD (g)} \times \text{population in each month (per 1,000 inhabitants)} \times 365 \text{ (days)}}$$

Japanese population data were obtained from the MHLW's *Annual Report on Population Dynamics* [16]. We identified drugs as branded or generic based on the MHLW drug classifications [17].

## Analysis

**Effect of incomplete data coverage in C-AMU.**   To clarify the effect of incomplete coverage in the claims database until 2014, we evaluated the longitudinal trends in S-AMU and C-AMU. First, we calculated monthly AMU from January 2013 to December 2017, and then we calculated the monthly ratio of S-AMU to C-AMU (SCR). We created boxplots of monthly SCR for each year and compared them using the Kruskal-Wallis test. Next, we analyzed the annual differences in SCRs from 2013, to 2015 and 2015 to 2017 using the Kruskal-Wallis test with Bonferroni correction for multiple comparisons.

**Effect of direct sales by pharmaceutical companies to medical facilities.**   As mentioned in the Introduction, drugs that are sold directly by pharmaceutical companies to medical facilities are not recorded in the S-AMU dataset. Because such patterns of sales are often seen for generic drugs, we compared the values of branded and generic drugs in S-AMU and C-AMU to ascertain the effect of direct sales by pharmaceutical companies to medical facilities. We calculated oral and parenteral drugs separately, and to remove the effects of the incomplete data coverage in C-AMU, we used data only from 2015 to 2017 in this analysis. After the descriptive evaluation of temporal trends in CSRs according to generic and branded drugs, we created boxplots of monthly SCR for 3 years (2015–2017) and compared them using the Mann-Whitney U test.

**Relationship between S-AMU and C-AMU.**   Finally, we evaluated the relationship between S-AMU and C-AMU in terms of generic versus branded drugs and oral versus parenteral drugs. In this analysis as well, we also used data only from 2015 to 2018. The relationship was assessed by linear regression, using coefficients and adjusted coefficients of determination (adjusted R squared). *P* values below 0.05 were considered statistically significant. All statistical analyses were performed using R ver. 4.0.2 (R Foundation for Statistical Computing, Vienna, Austria).

## Ethical approval

This study did not involve any interventions in human participants. All IQVIA and NDB data were fully anonymized before being sent to the authors. This study was approved by the institutional review board of the National Center for Global Health and Medicine (Approval Number: NCGM-G-002505-00).

## Results

### Effects of incomplete data coverage in C-AMU

Monthly DID values according to S-AMU and C-AMU over the study period are shown in **Table 1**, and temporal trends in SCRs are shown in **Fig 1**. The median and quartiles of annual

**Table 1. Temporal trends in national antimicrobial use according to sales and claims data from 2013 to 2017.**

|  | 2013 | | 2014 | | 2015 | | 2016 | | 2017 | |
|---|---|---|---|---|---|---|---|---|---|---|
|  | Sales | Claims | Sales | Claims | Sales | Claims | Sales | Claims | Sales | Claims |
| Jan | 15.52 | 15.21 | 14.36 | 14.76 | 14.61 | 15.81 | 13.55 | 14.75 | 14.02 | 15.34 |
| Feb | 15.90 | 15.23 | 14.66 | 15.36 | 13.83 | 14.95 | 15.74 | 16.97 | 14.46 | 15.20 |
| Mar | 13.20 | 14.17 | 14.25 | 14.74 | 14.09 | 15.21 | 13.53 | 15.57 | 13.62 | 14.44 |
| Apr | 15.47 | 14.76 | 14.99 | 14.52 | 15.45 | 15.49 | 15.67 | 14.73 | 13.69 | 13.63 |
| May | 15.24 | 14.42 | 13.52 | 13.88 | 13.20 | 14.05 | 13.66 | 13.75 | 13.58 | 14.19 |
| Jun | 13.07 | 12.74 | 12.82 | 12.80 | 13.87 | 14.28 | 13.24 | 13.59 | 13.20 | 13.56 |
| Jul | 13.58 | 12.94 | 13.05 | 12.91 | 13.74 | 13.82 | 12.79 | 13.42 | 11.96 | 12.45 |
| Aug | 12.35 | 11.86 | 11.51 | 11.73 | 12.02 | 12.40 | 12.92 | 12.84 | 12.08 | 12.07 |
| Sep | 13.37 | 12.87 | 13.70 | 13.54 | 14.49 | 14.49 | 13.32 | 13.37 | 12.51 | 13.05 |
| Oct | 15.55 | 14.61 | 15.69 | 14.93 | 16.09 | 16.29 | 15.46 | 15.78 | 14.72 | 14.53 |
| Nov | 17.22 | 16.34 | 15.19 | 15.43 | 16.30 | 16.07 | 17.20 | 17.02 | 15.45 | 15.24 |
| Dec | 18.59 | 16.22 | 20.03 | 17.64 | 18.44 | 17.66 | 18.27 | 17.46 | 16.45 | 15.83 |

Numbers represent defined daily doses per 1,000 inhabitants per day.

SCRs from 2013 to 2017 were 1.046 [1.036, 1.055], 0.993 [0.974, 1.017], 0.980 [0.936, 0.998], 0.987 [0.947, 1.007], 0.967 [0.956, 1.007], respectively. SCRs were highest in 2013, meaning that S-AMU was higher than C-AMU. Then, SCRs decreased before stabilizing after 2015. Differences in SCRs over 5 years were significant (Kruskal-Wallis test, $P = 0.002$). Furthermore, there were significant differences in SCRs among the years 2013, 2014, and 2015, even after Bonferroni correction (Kruskal-Wallis test, $P = 0.002$). Meanwhile, significant differences were not observed among the years 2015, 2016, and 2017. (Kruskal-Wallis test, $P = 0.926$). Fig 1A shows that the SCR for most months was less than 1 from 2015, meaning that C-AMU was higher than S-AMU.

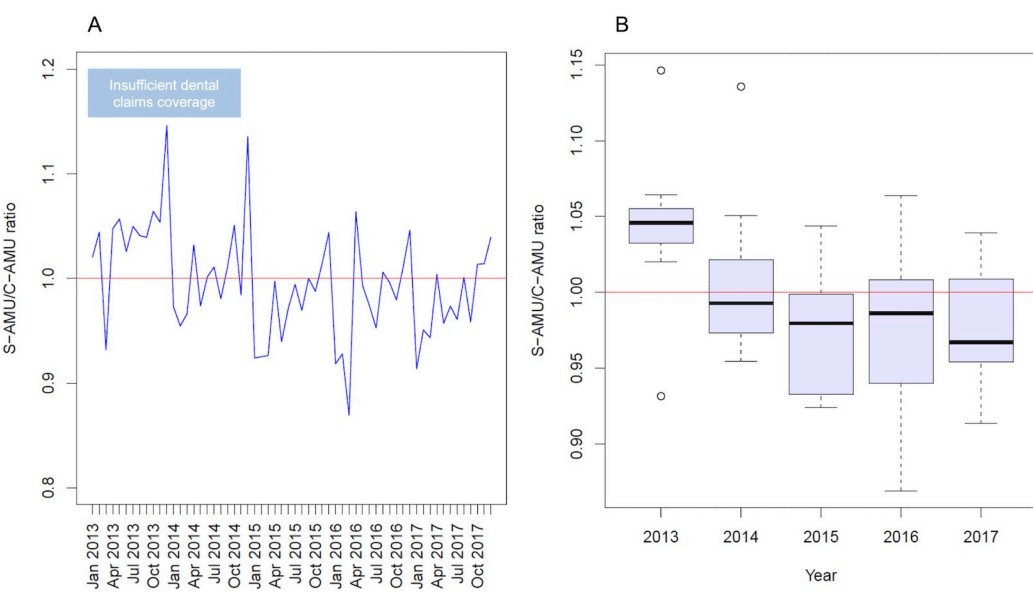

**Fig 1. Trends in the ratio of S-AMU to C-AMU from 2013 to 2017.** The line graph (Panel A) shows the temporal trends in SCRs. The boxplots (Panel B) show the annual SCRs. SCR was highest in 2013, then decreased and stabilized after 2015. Difference in SCRs over 5 years were significant ($P = 0.002$). SCRs were significantly different among the years 2013, 2014, and 2015 ($P = 0.002$), but not among the years 2015, 2016, and 2017. ($P = 0.926$).

**Table 2. Temporal trends in generic and branded antimicrobial use according to sales and claims data from 2015 to 2017.**

| Oral | 2015 | | | | 2016 | | | | 2017 | | | |
|---|---|---|---|---|---|---|---|---|---|---|---|---|
| | Generic | | Branded | | Generic | | Branded | | Generic | | Branded | |
| | Sales | Claims | Sales | Claims | Sales | Claims | Sales | Claims | Sales | Claims | Sales | Claims |
| Jan | 3.60 | 5.30 | 10.23 | 9.63 | 4.36 | 5.74 | 8.43 | 8.15 | 5.49 | 6.96 | 7.73 | 7.51 |
| Feb | 3.60 | 5.09 | 9.28 | 9.01 | 5.26 | 6.76 | 9.47 | 9.32 | 5.63 | 6.90 | 7.79 | 7.41 |
| Mar | 3.66 | 5.25 | 9.51 | 9.13 | 4.83 | 6.35 | 7.76 | 8.37 | 5.25 | 6.55 | 7.35 | 7.02 |
| Apr | 4.23 | 5.41 | 10.01 | 9.22 | 5.52 | 6.20 | 8.95 | 7.69 | 5.36 | 6.19 | 7.18 | 6.59 |
| Mar | 3.62 | 4.91 | 8.77 | 8.31 | 4.84 | 5.82 | 7.95 | 7.11 | 5.32 | 6.47 | 7.32 | 6.86 |
| Jun | 3.87 | 5.05 | 9.02 | 8.38 | 4.76 | 5.78 | 7.48 | 6.96 | 5.16 | 6.18 | 6.98 | 6.51 |
| Jul | 4.00 | 4.94 | 8.68 | 8.03 | 4.65 | 5.73 | 7.14 | 6.84 | 4.75 | 5.66 | 6.20 | 5.93 |
| Aug | 3.52 | 4.46 | 7.53 | 7.09 | 4.69 | 5.52 | 7.15 | 6.44 | 4.80 | 5.55 | 6.19 | 5.64 |
| Sep | 4.24 | 5.32 | 9.21 | 8.32 | 4.83 | 5.80 | 7.41 | 6.70 | 5.05 | 6.05 | 6.42 | 6.14 |
| Oct | 4.89 | 6.08 | 10.18 | 9.34 | 5.80 | 6.93 | 8.64 | 7.97 | 6.05 | 6.85 | 7.63 | 6.82 |
| Nov | 4.95 | 6.06 | 10.36 | 9.16 | 6.42 | 7.59 | 9.70 | 8.55 | 6.37 | 7.28 | 8.02 | 7.10 |
| Dec | 5.64 | 6.75 | 11.50 | 10.04 | 6.77 | 7.84 | 10.17 | 8.75 | 6.76 | 7.59 | 8.36 | 7.37 |
| Parenteral | 2015 | | | | 2016 | | | | 2017 | | | |
| Jan | 0.41 | 0.46 | 0.37 | 0.42 | 0.44 | 0.49 | 0.32 | 0.37 | 0.52 | 0.56 | 0.29 | 0.32 |
| Feb | 0.50 | 0.45 | 0.45 | 0.40 | 0.60 | 0.51 | 0.42 | 0.37 | 0.68 | 0.58 | 0.36 | 0.32 |
| Mar | 0.49 | 0.45 | 0.43 | 0.39 | 0.57 | 0.50 | 0.37 | 0.35 | 0.68 | 0.56 | 0.34 | 0.30 |
| Apr | 0.65 | 0.46 | 0.56 | 0.40 | 0.72 | 0.50 | 0.48 | 0.33 | 0.77 | 0.56 | 0.38 | 0.29 |
| Mar | 0.44 | 0.45 | 0.38 | 0.38 | 0.52 | 0.49 | 0.34 | 0.32 | 0.63 | 0.57 | 0.31 | 0.29 |
| Jun | 0.53 | 0.46 | 0.45 | 0.38 | 0.62 | 0.52 | 0.38 | 0.33 | 0.72 | 0.58 | 0.34 | 0.28 |
| Jul | 0.58 | 0.47 | 0.47 | 0.38 | 0.62 | 0.53 | 0.38 | 0.33 | 0.69 | 0.58 | 0.32 | 0.28 |
| Aug | 0.54 | 0.47 | 0.44 | 0.38 | 0.68 | 0.55 | 0.41 | 0.33 | 0.75 | 0.60 | 0.34 | 0.27 |
| Sep | 0.58 | 0.47 | 0.46 | 0.38 | 0.68 | 0.54 | 0.40 | 0.33 | 0.72 | 0.59 | 0.32 | 0.27 |
| Oct | 0.57 | 0.47 | 0.45 | 0.38 | 0.64 | 0.55 | 0.38 | 0.33 | 0.72 | 0.59 | 0.32 | 0.27 |
| Nov | 0.55 | 0.47 | 0.44 | 0.37 | 0.69 | 0.55 | 0.39 | 0.33 | 0.75 | 0.60 | 0.32 | 0.27 |
| Dec | 0.73 | 0.48 | 0.57 | 0.38 | 0.85 | 0.55 | 0.48 | 0.32 | 0.94 | 0.60 | 0.40 | 0.26 |

Numbers represent defined daily doses per 1,000 inhabitants per day.

## Effect of direct sales by pharmaceutical companies to medical facilities

Monthly DID values of generic and branded drugs according to S-AMU and C-AMU from 2015 to 20017 are shown in **Table 2**. The median and quartiles of oral generic, oral branded, parenteral generic, parenteral branded drug SCRs from 2015 to 2017 were 0.820 [0.787, 0.841], 1.079 [1.047, 1.112], 1.200 [1.139, 1.248], and 1.165 [1.121, 1.228], respectively. Temporal trends in SCR according to generic and branded drugs from 2015 to 2017 are shown in **Fig 2**. For oral drugs, the median SCRs of branded drugs were higher than 1, but those of generic drugs were lower than 1. This means that C-AMU was consistently higher than S-AMU for generic drugs. Meanwhile, in parenteral drugs, the SCR values of generic and branded drugs showed nearly the same values. SCRs of oral generic drugs increased from 2015 to 2017. Their median SCRs were consistently higher than 1. A significant difference was observed between the values of generic and branded drugs for oral drugs (Mann-Whitney U test, $P < 0.001$), whereas there was no apparent difference in parenteral drugs (Mann-Whitney U test, $P = 0.183$).

## Relationship between S-AMU and C-AMU

The relationship between S-AMU and C-AMU is illustrated using dot plots in **Fig 3**. For oral drugs, the coefficients of linear regression for generic and branded drugs were 0.90

Temporal trends in S-AMU/C-AMU ratio in terms of generic vs. branded drugs

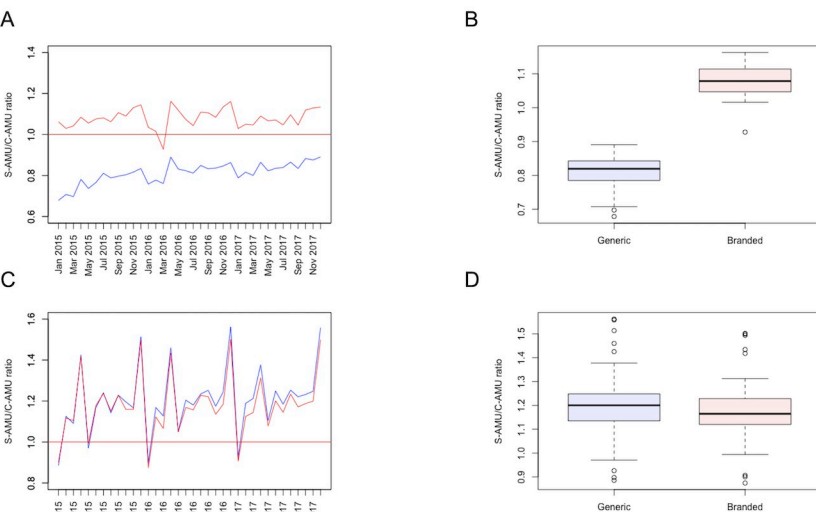

**Fig 2. Temporal trends in S-AMU/C-AMU ratio according to generic or branded drugs.** Panels A and B show line graphs of SCR trends in oral drugs and parenteral drugs, respectively. The median SCRs of oral branded drugs were higher than 1, but those of generic drugs were lower than 1. Panels C and D show boxplots of monthly SCRs from 2015 to 2017. A significant difference ($P < 0.001$) was observed in oral drugs, but not in parenteral drugs ($P = 0.183$).

($P < 0.001$) and 0.84 ($P < 0.001$), respectively; adjusted R squared values were 0.92 and 0.91, respectively. Meanwhile, for parenteral drugs, the coefficients of linear regression for generic and branded drugs were 0.32 ($P < 0.001$) and 0.43 ($P < 0.001$), respectively; adjusted R squared values were 0.52 and 0.40, respectively. Although the coefficients for both generic and branded drugs were significant in terms of both oral and parenteral drugs, the coefficients and adjusted R squared values were lower in parenteral drugs than in oral drugs.

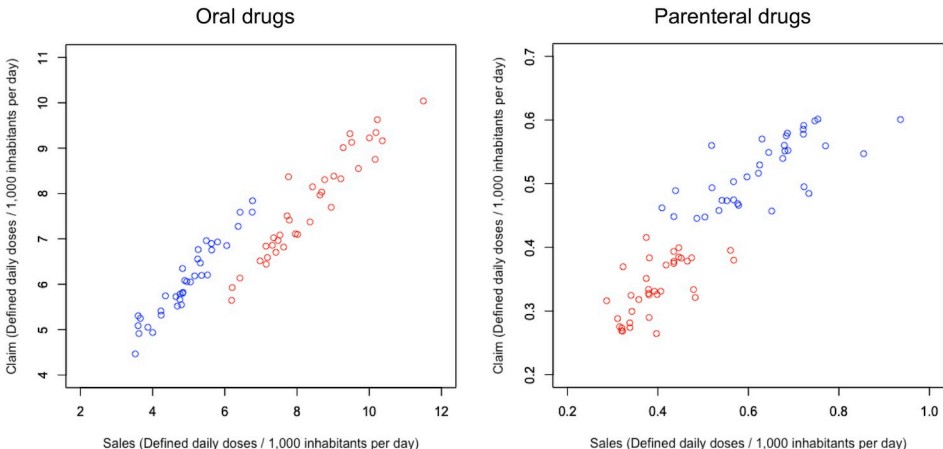

**Fig 3. Relationship between S-AMU and C-AMU in terms of oral vs. parenteral drugs.** Coefficients (*P* value) and adjusted R squared values for oral drugs were 0.90 ($P < 0.001$), 0.92 in generic drugs, and 0.84 ($P < 0.001$), 0.91 in branded drugs, and for parenteral drugs, were 0.32 ($P < 0.001$), 0.43 in generic drugs, and 0.52 ($P < 0.001$), 0.40 in branded drugs.

## Discussion

In this retrospective study, we calculated Japan's national S-AMU and C-AMU from 2013 to 2017 and clarified their characteristics and limitations. Currently, most national AMU surveillance is conducted using S-AMU or C-AMU, but few studies have assessed their characteristics. Hence, the results of this study provide insights into the sources of national AMU surveillance as well as its limitations.

In C-AMU, the proportion of electronically submitted claims is a key issue for data coverage. SCR was highest in 2013, decreased until 2015, and then stabilized. This finding reflects the incomplete coverage of the claims database due to the incomplete adoption of electronic claims submission in Japan until 2015, after which the coverage stabilized. Thus, we should consider this limitation when using C-AMU for AMU surveillance prior to 2015.

Direct sales by pharmaceutical companies to medical facilities are not recorded in the S-AMU dataset. The difference in SCRs between branded and generic drugs reflect this limitation of the sales data. It also explains why C-AMU is higher than S-AMU, even though S-AMU includes dead or disposed stock. This difference was observed for only oral drugs, suggesting that direct sales of parenteral drugs were not common. SCRs for generic oral drugs were increased throughout the study period, implying that direct sales have gradually diminished. We should monitor this temporal trend carefully to assess the national AMU surveillance based on S-AMU.

For both generic and branded drugs, the linear regression coefficients were close to 1 for oral drugs. Meanwhile, the coefficients were relatively low for parenteral drugs. This tendency was also observed in our previous study [8]. We attribute this to that healthcare claims being submitted per prescribed unit for oral drugs and per applied unit for parenteral drugs in the Japanese health insurance system. Hence, parenteral drugs that are canceled for clinical reasons such as cessation or change of antimicrobials might result in a discrepancy between S-AMU and C-AMU. In addition, since the DID of parenteral drugs is less than one-tenth that of the oral drugs in Japan [11, 12, 18], this discrepancy hardly affects the correlation between total S-AMU and C-AMU. However, we should consider this discrepancy when we assess only parenteral AMU.

These findings indicate that we can use both S-AMU and C-AMU to monitor national AMU. Therefore, when these characteristics and limitations are sufficiently understood, we can select the surveillance method according to the target purpose. For example, our findings indicate that S-AMU has utility as an alternative to C-AMU for monitoring temporal trends in national AMU [18]. This insight justifies our strategy of initially publishing a preliminary AMU report using S-AMU followed by a more comprehensive report using C-AMU, the latter of which includes more detailed information such as stratifications by patient age and sex. Likewise, it justifies the use of S-AMU for the Japanese government's Nippon AMR One Health Report. This report describes the aggregate AMU in both humans and animals with an aim toward adopting a "One Health" approach, which includes monitoring AMU not only in humans but also in animals [19].

Our study has several limitations. First, the analysis provides insight into S-AMU and C-AMU in Japan, but the characteristics and limitations of the data sources may not be directly applicable to those of other countries. For example, other countries might have a non-negligible proportion of antimicrobial sales without prescriptions [20, 21], which could lead to underestimations of C-AMU. In contrast, antimicrobial sales without prescriptions are not an issue in Japan due to strict regulations and easy access to medical facilities. Second, information on patients outside the national health insurance system is not available in the C-AMU. However, this may not greatly affect the longitudinal trends in C-AMU given that the number of such patients is likely to be consistent over time.

## Conclusion

This study comparatively explored differences in Japan's national AMU estimates based on sales and claims data. The data coverage was incomplete for C-AMU until 2015 due to insufficient adoption of electronically submitted claims, but the coverage was stabilized from 2015. Although S-AMU does not contain direct drug sales from pharmaceutical companies to hospitals, S-AMU and C-AMU are well-correlated. Therefore, we can use both datasets depending on the situation. An understanding of the characteristics of each data source should facilitate more accurate interpretations of national AMU estimates. Understanding the data source used for estimating AMU as well as its characteristics and limitations is desirable in every country.

## Author Contributions

**Conceptualization:** Yoshiki Kusama, Norio Ohmagari.

**Data curation:** Yoshiki Kusama, Yuichi Muraki, Chika Tanaka, Ryuji Koizumi, Masahiro Ishikane, Daisuke Yamasaki, Masaki Tanabe.

**Funding acquisition:** Norio Ohmagari.

**Investigation:** Yoshiki Kusama, Yuichi Muraki, Chika Tanaka, Ryuji Koizumi, Masahiro Ishikane.

**Methodology:** Yoshiki Kusama, Chika Tanaka, Daisuke Yamasaki, Masaki Tanabe.

**Supervision:** Yuichi Muraki, Masahiro Ishikane, Daisuke Yamasaki, Masaki Tanabe, Norio Ohmagari.

**Validation:** Yuichi Muraki, Ryuji Koizumi.

**Visualization:** Yoshiki Kusama.

**Writing – original draft:** Yoshiki Kusama.

**Writing – review & editing:** Yuichi Muraki, Chika Tanaka, Ryuji Koizumi, Masahiro Ishikane, Daisuke Yamasaki, Masaki Tanabe, Norio Ohmagari.

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
