## [Decision Letter · Decision Letter 0]

20 Oct 2020

PONE-D-20-21601

Characterization of the differences in national antimicrobial use surveillance between sales data and claims data in Japan

PLOS ONE

Dear Dr. Kusama,

Thank you for submitting your manuscript to PLOS ONE. After careful consideration, we feel that it has merit but does not fully meet PLOS ONE’s publication criteria as it currently stands. Therefore, we invite you to submit a revised version of the manuscript that addresses the points raised during the review process.

All reviewers comments are pasted below. The reviewers agree there is merit and this is a valuable contribution, but I urge you to consider their comments and submit a revision.

We look forward to receiving your revised manuscript.

Kind regards,

Fernanda C. Dórea

Academic Editor

PLOS ONE

Journal Requirements:

2)  In your ethics statement in the Methods section and in the online submission form, please provide additional information about the data used in your retrospective study. Specifically, please ensure that you have discussed whether all data were fully anonymized before you accessed them and/or whether the IRB or ethics committee waived the requirement for informed consent.

3)  Please describe how another researcher could obtain the data from IQVIA Japan used in this study.

Reviewers' comments:

Reviewer's Responses to Questions

**Comments to the Author**

1. Is the manuscript technically sound, and do the data support the conclusions?

Reviewer #1: Yes

Reviewer #2: No

2. Has the statistical analysis been performed appropriately and rigorously? 

Reviewer #1: N/A

Reviewer #2: No

3. Have the authors made all data underlying the findings in their manuscript fully available?

Reviewer #1: No

Reviewer #2: No

4. Is the manuscript presented in an intelligible fashion and written in standard English?

Reviewer #1: Yes

Reviewer #2: Yes

5. Review Comments to the Author

Reviewer #1: This is a very interesting exploration of differences between two comprehensive, population based methods to observe antibotic consumption in Japan. The authors are to be commended for performing such analyses in order to understand the available data better and in order to be able to deal with limitations. It is an example for researchers in other nations to do likewise.

The differences found are all by all minimal, which could incite the authors to conclude that Japan has 2 pretty reliable sources to monitor antibiotic consumption. The article should clearly state that this is true for consumption in primary care (and not hospital care). It this would be incorrect it should be stated, and it should also be indicated why it is not possible to make the distiinction between primary care and hospital care dispensing and reimbursment.

Some differences were observed but not statistically evaluated. Some explanation why this did not happen might be added (too small differences, complex reversing time lines).

The exploration of the small diferences leads to hypotheses on potential sources of bias. Most of these hypotheses are plausible. For OTC sales of antibiotics in Japan, the authors states that it is illegal. That does not mean that it does not happens, even in Japan. Some indication about the actual practices in Japan, and some references to the international literature on this should be advisable. It seems plausible that this might have an influence on the observerd difference in sales and claims of cheap oral antibiotics (where the remark on the legal status of OTC antibiotics was placed).

Reviewer #2: The topic of the paper is of utmost importance when it comes to monitoring AMR. However, I’m under the impression that the work has been hastily prepared. The paper in my view lacks clarity and the methods does not appear sound, unless the authors are dealing with data that lack granularity.

In general, thru my read, I’m left wanting more. The sources of my disappointment are as follows:

- Lack of description of the types of variables or information that are captured by S-AMU and C-AMU

- Methods look at first glance proper including potential time-series correlations but with limited granular data and its understanding, the authors are left with lots of hypotheses with some sense of triangulation of information/data

- Emphasis of the authors for the accuracy of AMU data while I believe estimating consistency of the data is much more important: I believe that the verification or validation of the consistency of the reporting from one year to another is more essential. Please recall we mean to use AMU trends as trends to be compared with AMR trends.

- Indeed, comparison between C-AMU and S-AMU is interesting (if you have access to data, which is not the case here). Both systems could be used to monitor AMR trends as opposed to disqualifying C-AMU but is not capturing comprehensively the data.

- Perceived some confusion re definitions between accuracy versus precision and between accuracy and biases; e.g. unclear meaning of inadequate accuracy? What does incomplete coverage mean? and moreover incomplete coverage does not mean necessary biases. E.g. incomplete data can be precise if they are representative of a situation.

Specific comments:

Line 26: abstract will benefit from a reference to the reason for AMU surveillance: AMR surveillance i.e. why these AMU estimates are important?

Line 34: unclear - as written - why the lack of paper-based claims is a limitation

Line 36: the authors could explain why analyzing direct sales to medical facilities is important; and why medical facilities will only buy generics – how about private hospitals?

Line 37: unclear about the importance of comparison between generic and brand drugs?

Line 38: unclear significance of paper-based and electronic claims

In short, without any understanding on how it works in Japan, I’m having a hard time processing the rationality of the methods

Line 41 – 44: the authors share DID values and trends. How do we know that these values are statistically significantly different between each other?

Line 46: interesting that DID of brand drugs are higher than generic drugs. Reasons are not explained, and values are not shown

Line 47: what is a dispensing claim

Line 51: unclear about the relationship between “AMU is underestimated” and “ lack of direct sales”; do the authors mean the data are missing or not analyzable/accessible?

Line 51: Conclusion of abstract is weak: lots of results displayed but no indication in the conclusion to make sense of them. E.g. last sentence is a kind of catchall phrase

Line 73: the authors argue that previous data did not account for time-series correlations. However, it appears the present study does not show any time-series correlation analysis either, does it?

Line 84: this paragraph on data sources should be expanded to insert much more information on Japan’s healthcare systems, detailed description of what is available in terms of variables and time periods of C-AMU and S-AMU. E.g. Private vs public hospitals; vet antibiotics versus antibiotics for humans; only one private company supplying Japan?

Line 105: meaning of inadequate accuracy? Is it important that the data is inaccurate if on can demonstrate that the data is precise and more importantly consistent overtime? If the authors claim inaccuracy – then what is the extent of the inaccuracy?

Line 106: just so that I’m clear: S-AMU data pertain to 99% of total sales but direct sales data to medical facilities are missing or not broken down or categorized as such in the dataset?

Line 117: would be interesting to mention how different these data are different from that of Europe in terms of available variables

Line 120: what is the extent of the AMU among tourists and for cosmetic procedures? Does it matter if we consider the inaccuracy is consistent over time? Recall that trends matter more for AMU and AMR surveillance purposes

Line 127: what kind of data is shared by IQVIA? Are there any limitations in relation to data sharing? Please detail what is shared by the company and what is available?

Results:

Line 153: again comparison between the 2 AMU data could be pertinent if these DID values are statistically significant or not?

Line 169: IQVIA accounts for 99% of the sales – is this for antimicrobials or all drugs? if data is made available, wouldn’t it be interesting to compare sales of antibiotics with that of other drugs, trends wise? Do you mean IQVIA has the monopoly of sales?

Line 170: Cant we work out some collaboration with the company for public health purposes to have access to more granular data?

Line 175-176: the main result in my view is the consistency of differences over many of these factors (parental vs per os, generic/branded). Shouldn’t the authors emphasize this more in the results?

Line 181 – 182: 2 sentences that say the same thing

Line 185: figure 2 seems to show interesting trends over time but no analysis?

Line 198: only one seems to realize that paper-based claims are not submitted into the system.

Line 220 -221: not sure I understand what sort of impact the authors were referring to

Line 222: difficult to understand this with limited background on how antimicrobials are supplied and distributed

Line 230: the paper becomes much more interesting if there is some attempt to estimate the extent of the biases. Unfortunately, we don’t have this in this paper

Line 245: I disagree with authors; C-AMU is valuable if consistent overtime

Line 247: this information could be shared earlier in a potential section on detailed description of AMU data

Line 260: why not doing the analysis of unadjusted DDD in this paper as this paper contains limited information and results

6. PLOS authors have the option to publish the peer review history of their article (what does this mean?). If published, this will include your full peer review and any attached files.

Reviewer #1: **Yes: **ROBERT Vander Stichele

Reviewer #2: No

---

## [Author Response · Author response to Decision Letter 0]

16 Dec 2020

Response to Reviewers

We have made major revisions to the entire manuscript based on the reviewers’ advice and suggestions. We are grateful for the opportunity to improve our work, and look forward to the reviewers’ evaluation of the new structure and theories included in the manuscript. Our point-by-point responses to the comments are provided below:

Reviewer 1

This is a very interesting exploration of differences between two comprehensive, population-based methods to observe antibiotic consumption in Japan. The authors are to be commended for performing such analyses in order to understand the available data better and in order to be able to deal with limitations. It is an example for researchers in other nations to do likewise.

Thank you for your time and effort in reviewing our work, and for your positive evaluation of our manuscript. We have made major revisions in accordance with your comments and suggestions.

1. The differences found are all by all minimal, which could incite the authors to conclude that Japan has 2 pretty reliable sources to monitor antibiotic consumption. The article should clearly state that this is true for consumption in primary care (and not hospital care). It this would be incorrect it should be stated, and it should also be indicated why it is not possible to make the distinction between primary care and hospital care dispensing and reimbursement.

Thank you for the advice. S-AMU and C-AMU include both primary care data and hospital data. However, S-AMU does not contain information about where an antibiotic is used, and therefore does not allow us to distinguish between primary care and hospital care dispensing/reimbursement. As advised, we have added the following information to the Data sources subsection of the Methods.

Page 7 Lines 106-109

However, S-AMU data do not include information on the purchasing parties (i.e., names and types of medical facilities) or patients, and therefore do not allow distinctions to be made between primary care and hospital care dispensing and reimbursement.

2. Some differences were observed but not statistically evaluated. Some explanation why this did not happen might be added (too small differences, complex reversing time lines).

As advised, we have added more statistical evaluations to our manuscript after consulting with a statistician.

Page 14 Lines 206-212

Effects on parenteral AMU

For parenteral antimicrobials, both branded and generic drugs exhibited similar trends in S-AMU and C-AMU (Fig 2A). The use of generic drugs increased while the use of branded drugs decreased over the study period. Furthermore, the scatter plots of branded and generic drugs demonstrated significant correlations between S-AMU (r = .100, P < .001) and C-AMU (r = .998, P < .001) (Fig 2B). For both branded and generic drugs, S-AMU was higher than C-AMU throughout the study period.

Page 15 Lines 223-232

Effects on oral AMU

The DID values of oral antimicrobials were more than 10 times higher than those of parenteral antimicrobials in both S-AMU and C-AMU. As with the parenteral antimicrobials, both branded and generic drugs in oral antimicrobials exhibited similar trends in S-AMU and C-AMU (Fig 3A). Again, the use of generic drugs increased while the use of branded drugs decreased over the study period. Throughout this period, S-AMU was higher than C-AMU for branded drugs, but C-AMU was higher than S-AMU for generic drugs. The scatter plots of branded and generic drugs demonstrated significant correlations between S-AMU (r = .996, P < .001,) and C-AMU (r = .997, P < .001) (Fig 3B).

3. The exploration of the small differences leads to hypotheses on potential sources of bias. Most of these hypotheses are plausible. For OTC sales of antibiotics in Japan, the authors states that it is illegal. That does not mean that it does not happens, even in Japan. Some indication about the actual practices in Japan, and some references to the international literature on this should be advisable. It seems plausible that this might have an influence on the observed difference in sales and claims of cheap oral antibiotics (where the remark on the legal status of OTC antibiotics was placed).

Thank you for suggesting this important point. OTC drugs are strictly regulated in Japan, and OTC antibiotics are, for all intents and purposes, not available (excluding some topical preparations). We would be hard pressed to think of any situation where a pharmacy would risk losing their retailing license and face criminal charges for illegally selling antibiotics that are easily and cheaply available with a physician’s prescription. In addition, physician consultations/prescriptions are highly affordable under Japan’s universal healthcare system. We have searched the international literature on the plausibility of OTC AMU in Japan, but could not find any relevant papers. In addition to the strict legislation on the sale of antibiotics, we think the main reason for this being such a non-issue is simply that the barriers to legally obtaining antibiotics are too low for illegal sales to be worth pursuing.

Although a very small number of personal trades/sales of antimicrobials through websites may occur, such AMU would not be reflected in sales or claims data. We have added these explanations to the limitations as follows:

Page 20 Lines 317-324

First, the analysis provides insight into S-AMU and C-AMU in Japan, but the characteristics and limitations of these data sources may not be directly applicable to those of other countries. For example, other countries may have a non-negligible proportion of antimicrobial sales without prescriptions [23, 24], which can lead to underestimations of C-AMU. In contrast, the inclusion of over-the-counter antimicrobial sales is not an issue in Japan due to strict regulations and easy access to prescribed antimicrobials. In this way, our findings should be evaluated in the context of Japan’s healthcare system and AMU patterns.

We have also cited the following additional references.

23. Batista AD, A Rodrigues D, Figueiras A, Zapata-Cachafeiro M, Roque F, Herdeiro MT. Antibiotic Dispensation without a Prescription Worldwide: A Systematic Review. Antibiotics (Basel). 2020;9: E786.

24. Sakeena MHF, Bennett AA, McLachlan AJ. Non-prescription sales of antimicrobial agents at community pharmacies in developing countries: a systematic review. Int J Antimicrob Agents. 2018;52: 771–82. 

Reviewer 2

The topic of the paper is of utmost importance when it comes to monitoring AMR. However, I’m under the impression that the work has been hastily prepared. The paper in my view lacks clarity and the methods does not appear sound, unless the authors are dealing with data that lack granularity. In general, thru my read, I’m left wanting more. The sources of my disappointment are as follows:

Thank you for your time and effort in reviewing our work, and we appreciate your advice and comments. The manuscript has undergone a major restructuring.

1. Lack of description of the types of variables or information that are captured by S-AMU and C-AMU

Thank you for your useful suggestion. For greater clarity, we have added explanations of S-AMU and C-AMU as follows:

S-AMU

Page 7, Lines 96-109

We estimated S-AMU using commercial data purchased from IQVIA Japan, a private data firm that curates databases of medical drug sales and distribution throughout the country. In Japan, medical drugs are generally sold by pharmaceutical companies to medical facilities via wholesalers. IQVIA Japan collects sales data from these wholesalers in order to construct databases. Although detailed information has not been published, the company states that it collects data from almost all domestic wholesalers, and that the data encompass more than 99% (monetary value) of all wholesale drug sales in Japan (this coverage was confirmed through personal communication with IQVIA Japan). Our dataset contained wholesalers’ sales data of all antimicrobials, as well as information on oral/parenteral classification and branded/generic classification. However, S-AMU data do not include information on the purchasing parties (i.e., names and types of medical facilities) or patients, and therefore do not allow distinctions to be made between primary care and hospital care dispensing and reimbursement.

C-AMU

Page 8, Lines 112-142

We estimated C-AMU using data obtained from the National Database of Health Insurance Claims and Specific Health Checkups of Japan (NDB), which is managed by the Ministry of Health, Labour and Welfare (MHLW). The NDB contains national-level health insurance claims data for healthcare encounters covered by insurance, and these data can be used for research purposes following the submission and approval of an application to the government [12]. Our dataset included data on the dates and types of antimicrobials prescribed to patients for all insurance-covered healthcare encounters.

In Japan, all residents are required to enroll in health insurance, which entitles them to receive insurance-covered healthcare at any medical facility throughout the country. Enrollees pay monthly premiums to their insurers, and also pay a fixed proportion (10–30% depending on age and income) of the medical charges at the point of care. Healthcare providers send claims to the applicable insurers to be reimbursed for the remaining charges. Because insurance-covered care accounts for the majority of medical treatments provided in Japan, the NDB represents a near-comprehensive database of all treatments performed throughout the country.

However, the NDB does not include claims data from patients with fully publicly funded healthcare (e.g., patients with intractable diseases, atomic bomb survivors, patients on welfare, patients with tuberculosis, and patients with human immunodeficiency virus infections) and patients who personally pay for all of their medical expenses (e.g., foreign travelers and cosmetic surgery patients). The detailed numbers of these patients are not published by the government.

Furthermore, the NDB only includes data from claims that have been submitted electronically, and the lack of paper-based claims reduces its coverage of healthcare encounters. Under Japan’s healthcare system, claims for medical care, dental care, and drug dispensing (by pharmacies) are handled separately. From 2011, electronically submitted claims for hospital-based medical care and drug dispensing claims have accounted for over 99.9% of total claims [13]. For clinic-based medical care, the proportions of electronically submitted claims increased from 91.0% in 2011 to 99.9% in 2017 [13]. For dental care, the proportions of electronically submitted claims have increased dramatically in recent years (31.5%, 46.4%, 55.7%, 69.5%, and 96.0% in 2011, 2012, 2013, 2014, and 2015, respectively) [13].

2. Methods look at first glance proper including potential time-series correlations but with limited granular data and its understanding, the authors are left with lots of hypotheses with some sense of triangulation of information/data

We agreed with your opinion, and have revised our study’s conceptual framework and development. Previously, we had, as you pointed out, aimed to explore the biases of S-AMU and C-AMU from various aspects. But we have changed this construction: We now consider the effects of the rapid shift in paper-based claims to electronic claims in dental care as a characteristic of C-AMU, and the effects of insufficient data coverage as a characteristic of S-AMU (which is not expected to greatly affect the representativeness of the AMU situation). This simplified the study, and allowed us to focus on the correlation between S-AMU and C-AMU. The specific changes are as follows:

Page 14, Lines 209-211

Furthermore, the scatter plots of branded and generic drugs demonstrated significant correlations between S-AMU (r = .100, P < .001) and C-AMU (r = .998, P < .001).

Page 15, Lines 230-232

The scatter plots of branded and generic drugs demonstrated significant correlations between S-AMU (r = .996, P < .001,) and C-AMU (r = .997, P < .001).

3. Emphasis of the authors for the accuracy of AMU data while I believe estimating consistency of the data is much more important: I believe that the verification or validation of the consistency of the reporting from one year to another is more essential. Please recall we mean to use AMU trends as trends to be compared with AMR trends.

Thank you for the important suggestion. As advised, we have examined the longitudinal consistency of S-AMU compared to C-AMU. The specific changes are as follows:

Page 10, Lines 159-162

To evaluate the longitudinal consistency of S-AMU, we compared the annual S-AMU with C-AMU between 2013 and 2017. In addition, we compared the use of branded and generic drugs in S-AMU and C-AMU to ascertain the effect of the government’s initiative to promote the shift to generic drugs.

4. Indeed, comparison between C-AMU and S-AMU is interesting (if you have access to data, which is not the case here). Both systems could be used to monitor AMR trends as opposed to disqualifying C-AMU but is not capturing comprehensively the data.

Thank you for the suggestion. We have examined the utility of S-AMU through a comparison with C-AMU in the revised manuscript. The specific changes are as follows:

Page 18, Lines 275-276

These findings indicate that S-AMU has utility as an alternative to C-AMU for monitoring the temporal trends in national AMU. 

5. Perceived some confusion re definitions between accuracy versus precision and between accuracy and biases; e.g. unclear meaning of inadequate accuracy? What does incomplete coverage mean? and moreover incomplete coverage does not mean necessary biases. E.g. incomplete data can be precise if they are representative of a situation.

Thank you for the comment. As mentioned in our response to Comment 3, we have focused the study on comparing S-AMU and C-AMU. In the revised manuscript, we consider the effects of the rapid shift in paper-based claims to electronic claims in dental care as a characteristic of C-AMU, and the effects of insufficient data coverage as a characteristic of S-AMU.

Specific comments:

6. Line 26: abstract will benefit from a reference to the reason for AMU surveillance: AMR surveillance i.e. why these AMU estimates are important?

We have clarified the importance of developing rigorous AMU surveillance systems as follows. 

Page 3, Lines 26-32

Purpose

Countries require rigorous antimicrobial use (AMU) surveillance systems to monitor prescription trends and evaluate the effects of antimicrobial resistance countermeasures. AMU can be estimated using sales data (S-AMU) or insurance claims data (C-AMU), but their differences are not well characterized. We comparatively examined Japan’s national S-AMU and C-AMU estimates across a 5-year period, and explored the potential causes of their differences.

7. Line 34: unclear - as written - why the lack of paper-based claims is a limitation

Thank you for the comment. The lack of paper-based claims potentially reduces the coverage of C-AMU. However, this may only have an effect on dental care (as reimbursements in the other types of care are predominantly managed by electronic claims). We have clarified this in the Data sources subsection of the Methods.

8. Line 36: the authors could explain why analyzing direct sales to medical facilities is important; and why medical facilities will only buy generics – how about private hospitals?

As we have set one of the study’s aims to clarify the effect of a government policy that encouraged the transition from branded drugs to generic drugs, the discussion about direct sales has become more important when compared to the previous manuscript. The description has been revised in the Discussion. In Japan, the government determines the drug prices for all administered/prescribed drugs under the national health insurance system. Therefore, patients are charged the same price for a specific drug regardless of where the drug was prescribed (public or private institutes).

9. Line 37: unclear about the importance of comparison between generic and brand drugs?

We added the importance of comparing the trends between branded and generic drugs. This was to clarify the effect of a government policy that encouraged the transition from branded drugs to generic drugs.

Page 6, Lines 82-85

In addition, longitudinal consistency is a crucial aspect of a national AMU surveillance system, but antimicrobial sales may be affected by changes to healthcare policies. For example, the Japanese government has encouraged the shift from branded drugs to generic drugs from 2013 onward [10, 11].

10. Line 38: unclear significance of paper-based and electronic claims. In short, without any understanding on how it works in Japan, I’m having a hard time processing the rationality of the methods

We apologize for the inadequate explanation. We have addressed the significance of paper-based and electronic claims in the additional explanation of C-AMU, as described in our response to Comment 1.

11. Line 41 – 44: the authors share DID values and trends. How do we know that these values are statistically significantly different between each other?

Because S-AMU and C-AMU are created from different data sources, it is difficult to interpret any statistical difference. Instead, we evaluated the correlation between S-AMU and C-AMU. Please refer to our response to Comment 2.

12. Line 46: interesting that DID of brand drugs are higher than generic drugs. Reasons are not explained, and values are not shown

The DID values of each group (parenteral/oral, branded/generic, and S-AMU/C-AMU) are shown in Table 1. The higher DID for branded drugs may be due to pharmaceutical companies promoting their sale over generic drugs to healthcare providers, or a large proportion of patients and doctors who choose branded drugs based on perceptions of higher effectiveness and safety. We have added a description of the government’s promotion of the transition from branded to generic drugs.

Page 6, Lines 84-85

For example, the Japanese government has encouraged the shift from branded drugs to generic drugs from 2013 onward [10, 11].

Page 18, Line 277-280

The initially high DID for branded drugs may have been due to their promotion by pharmaceutical companies and/or large proportions of patients and physicians who preferentially chose branded drugs based on perceptions of higher effectiveness and safety.

13. Line 47: what is a dispensing claim

A dispensing claim is an insurance claim for drug dispensing by pharmacies. We have clarified the language as follows.

Page 9, Lines 135-138

Under Japan’s healthcare system, claims for medical care, dental care, and drug dispensing (by pharmacies) are handled separately. From 2011, electronically submitted claims for hospital-based medical care and drug dispensing claims have accounted for over 99.9% of total claims [13].

14. Line 51: unclear about the relationship between “AMU is underestimated” and “lack of direct sales”; do the authors mean the data are missing or not analyzable/accessible?

The sentence has been removed from the revised manuscript.

15. Line 51: Conclusion of abstract is weak: lots of results displayed but no indication in the conclusion to make sense of them. E.g. last sentence is a kind of catchall phrase

We have revised the conclusion. It is difficult to meaningfully explain the results for branded/generic drugs and parenteral/oral drugs given the strict word limits of the abstract. Therefore, we have focused on the differences between S-AMU and C-AMU, and concluded that S-AMU may have utility in national AMU surveillance in Japan.

Page 4, Lines 51-56

Our study provides insight into the characteristics of S-AMU and C-AMU surveillance in Japan. Their differences may have been influenced by the omission of direct sales information in S-AMU and the lack of electronically submitted claims for dental care in C-AMU. However, these factors did not appear to differentially affect the temporal trends. The differences stabilized after 2015, suggesting that S-AMU is a viable surrogate indicator of Japan’s national AMU.

16. Line 73: the authors argue that previous data did not account for time-series correlations. However, it appears the present study does not show any time-series correlation analysis either, does it?

We have added an analysis of the correlation of S-AMU and C-AMU according to year. Please refer to our response to Comment 2.

17. Line 84: this paragraph on data sources should be expanded to insert much more information on Japan’s healthcare systems, detailed description of what is available in terms of variables and time periods of C-AMU and S-AMU. E.g. Private vs public hospitals; vet antibiotics versus antibiotics for humans; only one private company supplying Japan?

We have added more detailed explanations of S-AMU and C-AMU in our response to Comment 1. Furthermore, as advised, we have addressed veterinary AMU in the Discussion with newly cited reports.

Page 20, Lines 310-314

It is also important to adopt a “One Health” approach to controlling antimicrobial resistance, which includes monitoring AMU not only in humans, but also in animals [19]. The Japanese government’s “Nippon AMR One Health Report” described the use of S-AMU to calculate the aggregate AMU in both humans and animals [20].

18. Line 105: meaning of inadequate accuracy? Is it important that the data is inaccurate if on can demonstrate that the data is precise and more importantly consistent overtime? If the authors claim inaccuracy – then what is the extent of the inaccuracy?

We have made major revisions to the manuscript, and the sentences have been deleted.

19. Line 106: just so that I’m clear: S-AMU data pertain to 99% of total sales but direct sales data to medical facilities are missing or not broken down or categorized as such in the dataset?

We apologize for the inadequate explanation. S-AMU data included 99% of wholesale data. 

20. Line 117: would be interesting to mention how different these data are different from that of Europe in terms of available variables

We searched for lists of available variables of S-AMU in European countries, and contacted IQVIA about these variables. Unfortunately, we could not obtain any lists that could support such a comparison.

21. Line 120: what is the extent of the AMU among tourists and for cosmetic procedures? Does it matter if we consider the inaccuracy is consistent over time? Recall that trends matter more for AMU and AMR surveillance purposes

We agree with your comments. We have included a study aim to examine the longitudinal consistency between S-AMU and C-AMU, and addressed this lack of data from C-AMU in the limitations.

Page 20, Lines 324-327

Second, information on publicly funded and self-pay patients are not available in the C-AMU. However, this may not greatly affect the longitudinal trends in C-AMU as the number of such patients is likely to be consistent over time.

22. Line 127: what kind of data is shared by IQVIA? Are there any limitations in relation to data sharing? Please detail what is shared by the company and what is available?

We have added a more detailed explanation of the IQVIA data. The purchased data only contained sales of drugs from wholesalers to buyers (mainly medical facilities). IQVIA has restrictions prohibiting the public sharing of the data. However, researchers are free to purchase similar data directly from IQVIA.

23. Line 153: again comparison between the 2 AMU data could be pertinent if these DID values are statistically significant or not?

Thank you for your advice. We have examined the correlations between the 2 AMU data types as detailed in our response to Comment 2. 

24. Line 169: IQVIA accounts for 99% of the sales – is this for antimicrobials or all drugs? if data is made available, wouldn’t it be interesting to compare sales of antibiotics with that of other drugs, trends wise? Do you mean IQVIA has the monopoly of sales?

IQVIA accounts for 99% of all wholesale drug sales. Unfortunately, we had only purchased data regarding antimicrobials. While other vendors are free to collect and offer similar data, there are currently none that do so. 

25. Line 170: Cant we work out some collaboration with the company for public health purposes to have access to more granular data?

Thank you for your suggestion. IQVIA is unable to provide more granular data due to confidentiality agreements with the wholesalers.

26. Line 175-176: the main result in my view is the consistency of differences over many of these factors (parental vs per os, generic/branded). Shouldn’t the authors emphasize this more in the results?

Thank you for your suggestions. We have revised our study framework, and emphasize the consistency of the differences between generic vs branded drugs in both oral and parenteral drugs. Please refer to our response to Comment 2.

27. Line 181 – 182: 2 sentences that say the same thing

We have revised the sentence as follows:

Page 15, Lines 228-230

Throughout this period, S-AMU was higher than C-AMU for branded drugs, but C-AMU was higher than S-AMU for generic drugs.

28. Line 185: figure 2 seems to show interesting trends over time but no analysis?

We added Pearson’s correlation test to these results. Please refer to our response to Comment 2.

29. Line 198: only one seems to realize that paper-based claims are not submitted into the system.

We agree with your opinion. We have revised the discussion of paper-based claims such that it is now considered a characteristic of C-AMU.

30. Line 220 -221: not sure I understand what sort of impact the authors were referring to

This sentence was deleted as part of the major revisions to the manuscript.

31. Line 222: difficult to understand this with limited background on how antimicrobials are supplied and distributed

We have added the following explanation about this statement.

Page 18, Lines 285-289

In general, pharmaceutical companies sell their products to medical facilities through wholesalers to reduce the burden of sales. However, some companies may sell their products directly to medical facilities to increase their profit margins that would otherwise be lost to the wholesalers.

32. Line 230: the paper becomes much more interesting if there is some attempt to estimate the extent of the biases. Unfortunately, we don’t have this in this paper

Although we agree with this point, we were unable to quantify the extent of these biases with our data. Nevertheless, the study has shifted from speculating about these possible biases, and focused on the differences between C-AMU and S-AMU.

33. Line 245: I disagree with authors; C-AMU is valuable if consistent overtime

We have changed the discussion according to your advice, and regarded C-AMU as the standard method for evaluating the national AMU. Please refer to our response to Comment 4.

34. Line 247: this information could be shared earlier in a potential section on detailed description of AMU data

Thank you for the comment. However, the need for human-animal aggregate AMU is, although important, not central to our study aims. We have therefore kept this issue in the Discussion.

35. Line 260: why not doing the analysis of unadjusted DDD in this paper as this paper contains limited information and results

The reason for using population-adjusted DDD is written in the limitations. Adjusted DID is a more frequently used metric of national AMU than unadjusted DDD, with major international organizations (e.g., WHO and ECDC) using this unit to evaluate the AMU of countries. If readers would like to evaluate unadjusted DDD, they can easily obtain the figures by multiplying DID with the population of Japan, as described in Reference #15 (Statistics of Japan).

Page 21, Lines 327-332

Finally, we evaluated AMU using DDD values adjusted to the national population. Because longitudinal evaluations that use this metric are affected by population changes, unadjusted DDD estimates may represent a more suitable metric to evaluate AMU transitions. Nevertheless, national AMU evaluations frequently use adjusted DID as an indicator, which allows for more practical and intuitive interpretations than unadjusted values.

---

## [Decision Letter · Decision Letter 1]

2 Feb 2021

PONE-D-20-21601R1

Characterization of the differences in national antimicrobial use surveillance between sales data and claims data in Japan

PLOS ONE

Dear Dr. Kusama,

Thank you for submitting your manuscript to PLOS ONE. After careful consideration, we feel that it has merit but does not fully meet PLOS ONE’s publication criteria as it currently stands. Therefore, we invite you to submit a revised version of the manuscript that addresses the points raised during the review process.

I agree with the reviewers that the paper presents an important study. A major revision is recommended mainly on the basis of the need to perform and present more thorough statistical analysis. 

We look forward to receiving your revised manuscript.

Kind regards,

Fernanda C. Dórea

Academic Editor

PLOS ONE

Additional Editor Comments (if provided):

I would like to reinforce the opinion of both reviewers that the authors put significant effort into replying to the comments from the previous version, and a lot of improvements have been made. However, I concur with all of reviewer's 2 comments in this new version.

In particular:

1) The new to state more clearly what is the central research question of the paper, and present results as main finding, and then broken down into more detailed conclusions. If the main goal of the paper to investigate whether S-AMU data can be used as a surveillance tool as a substitute to C-AMU? Then needs to be stated clearly. Right now the abstract and introduction go back and forth between saying that the paper will look for differences between the two sources, and that the paper will look for similarities (correlation). These seem apparently contradicting unless clearly stated as parts of the overarching goal. Which is not to just compare the two, but specifically to see if S-AMU can be considered as good as C-AMU for the purpose of surveilling trends in usage.

2) The statistics needs much improvement. All statements made should be defended by statistical results (avoid statements like "consistently higher", and instead give statistical results and confidence).

Regarding 2), I question in particular why data were aggregated yearly, rather than for instance monthly. Doing correlation and trend analysis with 5 data points may never give the confidence you need to backup the conclusions you are trying to make.

Please pay particular attention to the reviewer's comments on the figures (there seems to be a contradiction between what the text says about the figures 2B and 3B, and what they actually show).

Please review the wording on the sentence in lines 64-66.

Reviewers' comments:

Reviewer's Responses to Questions

**Comments to the Author**

1. If the authors have adequately addressed your comments raised in a previous round of review and you feel that this manuscript is now acceptable for publication, you may indicate that here to bypass the “Comments to the Author” section, enter your conflict of interest statement in the “Confidential to Editor” section, and submit your "Accept" recommendation.

Reviewer #1: All comments have been addressed

Reviewer #2: All comments have been addressed

2. Is the manuscript technically sound, and do the data support the conclusions?

Reviewer #1: Yes

Reviewer #2: Partly

3. Has the statistical analysis been performed appropriately and rigorously? 

Reviewer #1: Yes

Reviewer #2: No

4. Have the authors made all data underlying the findings in their manuscript fully available?

Reviewer #1: Yes

Reviewer #2: Yes

5. Is the manuscript presented in an intelligible fashion and written in standard English?

Reviewer #1: Yes

Reviewer #2: No

6. Review Comments to the Author

Reviewer #1: The auhtors have responded well to the two reviewers' comments. There are no futher recommendations to be made.

Reviewer #2: I would like to congratulate the authors for carefully addressing the reviewers comments. the ms is now much clearer and I do understand the approach and the findings better.

However, in my view i would encourage to have a second look of the results and discussion sections. Results are in my opinion are not well presented with limited used of statistical tools e.g. trends and not values should be the focus; Discussion is confusing and not well presented. e.g. the authors should start with the major findigns, the major recommendations and break down into secondary findings.

Please find attached my specific comments

7. PLOS authors have the option to publish the peer review history of their article (what does this mean?). If published, this will include your full peer review and any attached files.

Reviewer #1: **Yes: **ROBERT Vander Stichele

Reviewer #2: No

---

## [Author Response · Author response to Decision Letter 1]

18 Mar 2021

Reviewer #1: The authors have responded well to the two reviewers' comments. There are no further recommendations to be made.

We sincerely appreciate your kind review.

Reviewer #2:

Thank you for reviewing our manuscript and providing helpful comments. We have thoroughly reconsidered the fundamental structure of the manuscript based on the editor and reviewer comments. In particular, we have clarified the objective of the study, namely, elucidating the characteristics and limitations of S-AMU and C-AMU. To evaluate these objectives, we created monthly S-AMU and C-AMU trends and performed corresponding statistical analyses.

The major revisions are as follows:

• We changed the observation time-scale from yearly to monthly.

• We evaluated the S-AMU/C-AMU ratio (SCR) to assess trends in the relationship between S-AMU and C-AMU.

• We removed the comparison of penicillin and fluoroquinolone because it was ambiguous.

• We changed the method of analysis from correlation analysis to linear regression for the comparison of S-AMU and C-AMU.

• We changed the title, abstract, and Discussion section, accordingly.

I would like to congratulate the authors for carefully addressing the reviewer’s comments. the ms is now much clearer and I do understand the approach and the findings better. However, in my view I would encourage to have a second look of the results and discussion sections. Results are in my opinion are not well presented with limited used of statistical tools e.g. trends and not values should be the focus; Discussion is confusing and not well presented. e.g. the authors should start with the major findings, the major recommendations and break down into secondary findings. Please find attached my specific comments.

Thank you for your meticulous suggestions. In accordance with your comments, we have reconsidered the fundamental structure of the manuscript and revised nearly all of it from the beginning. Our point-by-point responses are as follows.

Abstract

1. Line 46 what is the overall result? to me this is not the main finding. you can break down later.

We have presented our findings numerically and statistics-based terms to the extent possible in the revised manuscript. Please let us know if you have further concerns or suggestions.

2. Line 46 what does it mean - difference is correlated or trends is correlated?

We have clarified the statistical methods in both the abstract and the main text.

3. Line 47 where are the p values for these comparisons?

We have presented our findings numerically and statistics-based terms to the extent possible in the revised manuscript. Please let us know if you have further concerns or suggestions.

4. Line 51 overall results? It should be presented first.

We have revised the conclusion accordingly. Specifically, we now present the overall results descriptively, rather than numerically.

5. Line 55 do you mean that C-AMU is not good? why?

We did not intend to imply that C-AMU is not good. Accordingly, we have revised the description for clarification. Both types of AMU are useful depending on the situation.

The revised abstract is as follows.

Abstract

Purpose

Antimicrobial use (AMU) is estimated at the national level by using sales data (S-AMU) or insurance claims data (C-AMU). However, these data might be biased by generic drugs that are not sold through wholesalers (direct sales) or recorded in sales databases, or by claims that are not submitted electronically or stored in claims databases. We evaluated these effects by comparing S-AMU and C-AMU to ascertain the characteristics and limitations of each kind of data. We also evaluated the interchangeability of these data by assessing their relationship. 

Methods

We calculated monthly defined daily doses per 1,000 inhabitants per day (DID) using sales and claims data from 2013 to 2017. To assess the effects of non-electronic claim submissions on C-AMU, we evaluated trends in the S-AMU/C-AMU ratio (SCR). To assess the effects of direct sales of S-AMU, we divided AMU into generic and branded drugs and evaluated each SCR in terms of oral versus parenteral drugs. To assess the relationship between S-AMU and C-AMU, we created a linear regression and evaluated its coefficient.

Results

Median annual SCRs from 2013 to 2017 were 1.046, 0.993, 0.980, 0.987, and 0.967, respectively. SCRs dropped from 2013 to 2015, and then stabilized. Differences in SCRs between branded and generic drugs were significant for oral drugs (0.820 vs 1.079) but not parenteral drugs (1.200 vs 1.165), suggesting that direct sales of generic drugs were omitted in S-AMU. Coefficients of DID between S-AMU and C-AMU were high (generic, 0.90; branded, 0.84) in oral drugs but relatively low (generic, 0.32; branded, 0.52) in parenteral drugs.

Conclusions

The omission of direct sales information and non-electronically submitted claims have influenced S-AMU and C-AMU information, respectively. However, these data were well-correlated, and it is considered that both kinds of data are useful depending on the situation.

Instruction

6. Line 80 does this study have the same period as the study of the present ms?

We have added the study periods of the reference as follows.

Line 77-81

For C-AMU, the scope of medical care covered by health insurance is a key factor in the validity of the dataset. In Japan, nearly all medical care is covered by the national health insurance system, with a few exceptions such as cosmetic surgery and travel medicine (the number of items covered has not changed over time, according to the study conducted from 2011 to 2013 [8]).

Methods

7. Line 108 no distinction either regarding geographic locations of those facilities?

Although the geographic locations are registered in the database, we cannot use that information because of IQVIA Japan’s rules concerning data provision.

8. Line 146 please remind what J01 is about

We have added an explanation of J01 as follows.

Line 152-153

Antimicrobials are identified using the Anatomical Therapeutic Chemical (ATC) classification J01 (anti-infectives for systemic use)

9. Line 149 what is a maintenance dose

We have added an explanation of maintenance dose as follows.

Line 158 (i.e., not loading dose)

Results

10. Line 191 I don’t see a P correlation coef value and its p value?

We have added the following statistical information to this part.

Lines 211-218

Differences in SCRs over 5 years were significant (Kruskal-Wallis test, P = 0.002). Furthermore, there were significant differences in SCRs among the years 2013, 2014, and 2015, even after Bonferroni correction (Kruskal-Wallis test, P = 0.002). Meanwhile, significant differences were not observed among the years 2015, 2016, and 2017. (Kruskal-Wallis test, P = 0.926). Panel A of Figure 1 shows that the SCR for most months was less than 1 from 2015, meaning that C-AMU was higher than S-AMU.

11. Line 203 where are the comparison values and p values?

The tables now show the DID values (for replication studies), and the P values are now presented in the footnote of the figures.

Lines 227-231, Figure 1

Panel A shows the temporal trends in SCRs. Panel B shows the annual SCRs. SCR was highest in 2013, then decreased and stabilized after 2015. Difference in SCRs over 5 years were significant (P = 0.002). SCRs were significantly different among the years 2013, 2014, and 2015 (P = 0.002), but not among the years 2015, 2016, and 2017. (P = 0.926).

12. Line 211 I don’t understand these values - what are you comparing? and these values don’t appear in fig 2 either

We assessed the coefficient of the linear regression between S-AMU and C-AMU. The results of the statistical analysis are shown in the footnote of Figure 3.

Line 272, Figure 3

Coefficients (P value) and adjusted R squared values for oral drugs were 0.90 (P < 0.001), 0.92 in generic drugs, and 0.84 (P < 0.001), 0.91 in branded drugs, and for parenteral drugs, were 0.32 (P < 0.001), 0.43 in generic drugs, and 0.52 (P < 0.001), 0.40 in branded drugs.

13. Line 212 what is higher?

We have removed this sentence.

14. Line 218 sorry I dont understand the significance of it; if the authors want to correlate, it should be by year... using another type of graph

We consider that this issue has been resolved by the use of monthly plots.

15. Line 231 I understand now what these 2 values of coefficients refer to but not correctly presented - please revise not shown in figure 3 as well - they should

We have assessed the coefficients of the linear regression between S-AMU and C-AMU, as described in our response to comment #12. 

16. Line 244 please present these data in trends - very difficult to read adn compare trends - presenting ranges of values are to me secondary!

Thank you for this suggestion. We rewrote the manuscript in statistical terms to the extent possible. Accordingly, we judged that this figure (trends in DIDs of penicillin and fluoroquinolone) was no longer necessary and so it has been removed from the manuscript.

Discussion

17. Line 260 In my view, the authors should make an effort to present the discussion in a more classic way:

- start with the main finding and consequences (I had to wait for the conclusion to understand the main recommendations)

- other findings that are less secondary and limitations could come later

We have revised the Discussion section for flow and clarity.

18. Line 263 again, to me, this is not about values but trends: 

- trends differ during 2013-2015

- trends are the same over the next 3 years

would these trends be similar, both systems can be used interchangeably to monitor the impact of AMU over AMR trends... correct? For practicality reasons, we recommend C AMU as it is made available .... correct?

We now present these two points in statistical terms and have revised the relevant text for clarity.

19. Line 264 this sentence is not clear - please revise

The description has been revised along with the changes to the rest of the manuscript. I believe this part is now clearer than before.

20. Line 276 don’t understand - why not use both? do we need to choose?

Yes, we should use both types of data depending on the situation. We have revised this description as follows.

Lines 309-311

These findings indicate that we can use both S-AMU and C-AMU to monitor national AMU. Therefore, when these characteristics and limitations are sufficiently understood, we can select the surveillance method according to the target purpose.

21. Line 284 ?? posit?

We have deleted the word “posit”.

22. Line 300 this is the main finding

Thank you for noting this. We now present this as the main finding of the study.

---

## [Decision Letter · Decision Letter 2]

26 Apr 2021

Characteristics and limitations of national antimicrobial surveillance according to sales and claims data

PONE-D-20-21601R2

Dear Dr. Kusama,

We’re pleased to inform you that your manuscript has been judged scientifically suitable for publication and will be formally accepted for publication once it meets all outstanding technical requirements.

Kind regards,

Fernanda C. Dórea

Academic Editor

PLOS ONE

Additional Editor Comments (optional):

Reviewers' comments:

Reviewer's Responses to Questions

**Comments to the Author**

1. If the authors have adequately addressed your comments raised in a previous round of review and you feel that this manuscript is now acceptable for publication, you may indicate that here to bypass the “Comments to the Author” section, enter your conflict of interest statement in the “Confidential to Editor” section, and submit your "Accept" recommendation.

Reviewer #1: (No Response)

Reviewer #2: All comments have been addressed

2. Is the manuscript technically sound, and do the data support the conclusions?

Reviewer #1: (No Response)

Reviewer #2: Yes

3. Has the statistical analysis been performed appropriately and rigorously? 

Reviewer #1: (No Response)

Reviewer #2: Yes

4. Have the authors made all data underlying the findings in their manuscript fully available?

Reviewer #1: (No Response)

Reviewer #2: Yes

5. Is the manuscript presented in an intelligible fashion and written in standard English?

Reviewer #1: (No Response)

Reviewer #2: Yes

6. Review Comments to the Author

Reviewer #1: (No Response)

Reviewer #2: the authors have responded well to my comments. One final recommendations would be to convert figures 1 and 2 into graphs.

7. PLOS authors have the option to publish the peer review history of their article (what does this mean?). If published, this will include your full peer review and any attached files.

Reviewer #1: **Yes: **ROBERT Vander Stichele

Reviewer #2: **Yes: **Sirenda Vong

---

## [Editor Report · Acceptance letter]

29 Apr 2021

PONE-D-20-21601R2 

Characteristics and limitations of national antimicrobial surveillance according to sales and claims data 

Dear Dr. Kusama:

I'm pleased to inform you that your manuscript has been deemed suitable for publication in PLOS ONE. Congratulations! Your manuscript is now with our production department. 

Kind regards, 

on behalf of

Dr. Fernanda C. Dórea 

Academic Editor

PLOS ONE